# β2-Adrenergic Receptor Mediated Inhibition of T Cell Function and Its Implications for CAR-T Cell Therapy

**DOI:** 10.3390/ijms241612837

**Published:** 2023-08-16

**Authors:** Muhammad Asad Farooq, Iqra Ajmal, Xinhui Hui, Yiran Chen, Yaojun Ren, Wenzheng Jiang

**Affiliations:** Shanghai Key Laboratory of Regulatory Biology, School of Life Sciences, East China Normal University, Shanghai 200241, China; 52181300056@stu.ecnu.edu.cn (M.A.F.); 52171300059@stu.ecnu.edu.cn (I.A.);

**Keywords:** β2-adrenergic receptor, CAR-T therapy, tumor microenvironment, immunosuppression, immunotherapy

## Abstract

The microenvironment of most tumors is complex, comprising numerous aspects of immunosuppression. Several studies have indicated that the adrenergic system is vital for controlling immunological responses. In the context of the tumor microenvironment, nor-adrenaline (NA) is poured in by innervating nerves and tumor tissues itself. The receptors for nor-adrenaline are present on the surfaces of cancer and immune cells and are often involved in the activation of pro-tumoral signaling pathways. Beta2-adrenergic receptors (β2-ARs) are an emerging class of receptors that are capable of modulating the functioning of immune cells. β2-AR is reported to activate regulatory immune cells and inhibit effector immune cells. Blocking β2-AR increases activation, proliferation, and cytokine release of T lymphocytes. Moreover, β2-AR deficiency during metabolic reprogramming of T cells increases mitochondrial membrane potential and biogenesis. In the view of the available research data, the immunosuppressive role of β2-AR in T cells presents it as a targetable checkpoint in CAR-T cell therapies. In this review, we have abridged the contemporary knowledge about adrenergic-stress-mediated β2-AR activation on T lymphocytes inside tumor milieu.

## 1. Introduction 

Neurotransmitters play a bridging role between the nervous system and the body. Catecholamines are important set of neurotransmitters released by supra-renal glands and comprise adrenaline/epinephrine (A/E) and nor-adrenaline/nor-epinephrine (NA/NE). Adrenergic receptors are activated in response to stimulation by adrenaline and nor-adrenaline to chemically coordinate the signals/messages from nervous system to the target tissues [1,2]. Adrenergic receptors (ARs) belong to the G-protein coupled, seven transmembrane receptor (GPCR) family and constitute alpha-adrenergic receptor (α-AR) and beta-adrenergic receptor (β-AR) subtypes, which further have been classified as Alpha-1 adrenergic receptors (α1-ARs), Alpha-2 adrenergic receptors (α2-ARs), Beta-1 adrenergic receptors (β1-ARs), Beta-2 adrenergic receptors (β2-ARs) and Beta-3 adrenergic receptors (β3-ARs). In short-term or acute stress conditions like fear or exercise, the body releases catecholamines via activation of the sympathetic nervous system (SNS) that bind with adrenergic receptors on various organs, resulting in increased heart rate, dilation of pupils, mobilization of energy, and diversion of the blood flow from other body organs to the skeletal muscles to cope with the aforementioned stress [3].

However, in chronic stress conditions, the body retains a persistently higher concentration of catecholamines and sustained activation of SNS, leading to the initiation and progression of cancer [4]. Many studies have demonstrated that this cancer progression is due to catecholamines triggering β-adrenergic receptors, more specifically, downstream signaling of β2-Adrenergic receptors in cancer cells [5]. Later on, it was demonstrated that psychological factors and depressive disorders can lead to cancer incidence. During stress, elevated levels of catecholamines can lead to cancer initiation by making the genome unstable, rendering them exposed to environmental carcinogens [6]. Because of this, more studies were carried out aiming to unearth the role of β2-AR in tumor initiation and progression. Recently, the role of β2-AR in immunity has been the focus of many studies. β2-Adrenergic receptors can modulate the functions of various immune cells as the corresponding receptors are displayed on the surface of T cells, natural killer (NK) cells, and dendritic cells (DCs) [7,8]. The ligand for β2-AR is released inside the tumor microenvironment (TME) and leads to inhibitory signaling in T lymphocytes [9,10].

The immune system plays a critical role in identifying and eliminating cells that undergo malignant transformation, in addition to its primary function of eradicating infectious agents. If the inflammatory response remains unresolved, the affected cells undergo transformation and become cancerous [11]. However, at this stage, the immune system identifies the tumor cells by recognizing the tumor-specific antigens displayed on their surfaces [12], followed by effector immune responses primarily mediated by CD8+ T cells and NK cells. The activation of these immune responses forms the basis for immunotherapy, as it halts tumorigenesis [13,14].

Due to the remarkable success of the initial clinical trials of chimeric antigen receptor (CAR)-T therapy, particularly in pediatric patients with hematological malignancies, the clinical response rates in leukemia patients have reached as high as 90% [15,16]. As a result, the number of successful clinical trials for CAR-T therapy targeting hematological malignancies which direct numerous antigens has upsurged significantly, yet the parallel success of CAR-T therapy in solid tumors is still being awaited [17,18]. The number of ongoing clinical trials in solid tumors is far less than in liquid tumors, courtesy of the toxic side effects and suboptimal therapeutic outcomes which are achieved almost every time. There can be several explanations for this, including the following. Hematological cancers commonly exhibit similar antigens, and their distribution across various types of hematological cancers is generally similar, with some exceptions [19]. On the contrary, antigens related to solid tumors vary greatly, not just from tumor to tumor, but also between different forms of a similar tumor, i.e., primary and metastatic forms [20]. Moreover, hematological cancers are widespread in the circulatory system, which makes them less dense and easily targetable as compared to solid tumors. The solid tumors are denser and are concentrated on a single site, creating a physical barrier for CAR-T cells by developing extensive vasculature to supply ample nutrients to the fast-growing tumor, thus inhibiting the chemo-attractive signals that are necessary for CAR-T cells to reach the tumor site [21]. The immune-suppressive characteristics of solid tumors, which arise due to the presentation of checkpoint inhibitory ligands and metabolites from diverse metabolic pathways collectively create a tumor microenvironment (TME). This microenvironment makes it exceedingly challenging for CAR-T cells to infiltrate and effectively eliminate solid tumors, leading to significant barriers to CAR-T cell therapy’s success [22]. Adrenergic stress is one of the culprits, among others, responsible for immunosuppression inside the TME. Previously, we have attempted to target immunosuppressive factors inside the TME in an attempt to increase the efficacy of CAR-T therapy in the prostate and pancreatic tumor microenvironments [23,24].

The role of adrenergic stress in cancer and immunity has been reviewed previously [25,26,27]. In this review, our primary focus will be on exploring the ramifications of adrenergic stress-induced activation of β2-adrenergic receptors (β2-AR) in T lymphocytes and its potential impact on CAR-T cell therapy. To the best of our knowledge, the inhibitory role of β2-AR in T lymphocytes, particularly in the context of CAR-T therapy, remains a novel area of investigation with no prior publications. Our goal is to delve deeply into the mechanisms of immunosuppression within the tumor microenvironment (TME) and how it negatively modulates CAR-T signaling.

## 2. Nor-Adrenaline in Tumor Microenvironment

The production of adrenaline and nor-adrenaline within the TME can arise from various sources, including the tumor cells themselves, the nerve fibers that innervate the tumors, and the surrounding stromal cells such as fibroblasts and immune cells. It seems that NA, mainly from the innervating nerves, plays a role in the incidence and initial progression of the cancer, as TME is absent initially. After TME is established, it is mainly the TME that pours sufficient amounts of NA into the tumor surroundings that carry its immunosuppressive roles in CD8+ T cells. This NA comes mainly from two sources. Firstly, the tumor itself contains all the necessary machinery to synthesize NA. Secondly, many tumor types are innervated by sympathetic nerve fibers [28]. Neuronal progenitor cells migrate towards the tumor tissue during the process of neurogenesis and adapt a sympathetic tone while innervating tumor tissue. Moreover, the sensory neurons innervating tumor tissues are also reprogrammed to sympathetic nerve phenotypes [29]. Altogether, this creates hyperactive SNS signaling inside the TME. Growing evidence suggest that an immunosuppressive role of the adrenergic system inside the TME could possibly hinder the antitumor functions of immunotherapy [6,27].

## 3. Adrenergic Stress Endorses Immunosuppressive Tumor Milieu Development

The catecholamines bind to adrenergic receptors on target cells, including immune cells, and trigger a range of biological responses. In the context of the immune system, adrenergic stress can change the activities of different immune cell subsets due to its wide expression. One of the key mechanisms by which adrenergic stress promotes immunosuppression is through the recruitment and activation of myeloid-derived suppressor cells (MDSCs) [30]. MDSCs are regulatory immune cells that have the skill to inhibit T cells and promote tumor growth. Adrenergic stress has been shown to promote the expansion and activation of MDSCs in the tumor microenvironment through the activation of adrenergic receptors on MDSCs themselves, as well as on other cells that produce cytokines and chemokines which promote the recruitment and activation of MDSCs [31]. β2-AR activation on MDSCs has been shown to enhance the expression of arginase-1, an enzyme that can deplete arginine, an amino acid that is important for T cell function. Additionally, adrenergic stress inhibits T cell functions by upregulating interleukin-10 (IL-10) and transforming growth factor-beta (TGF-β) [32,33]. The production of these anti-inflammatory cytokines, owing to the activation of β2-AR on the surfaces of tumor-associated macrophages (TAMS), also demonstrated as M2-phenotype. Adrenergic signaling in cancer plays a significant role in fostering the release of vascular endothelial growth factor (VEGF), a crucial mediator of angiogenesis, from both tumor cells and M2-macrophages residing in the tumor microenvironment [34]. VEGF facilitates the recruitment of endothelial cells, leading to the formation of new blood vessels, which in turn support the delivery of oxygen and nutrients to the tumor [27]. Additionally, the activation of β2-adrenergic receptors (β2-AR) can enhance the activity of regulatory T cells (Tregs), thereby amplifying the immunosuppressive environment around the tumor [35,36].

According to the literature, adrenergic stress has been shown to inhibit T cell activation and proliferation, and to promote apoptosis. Similarly, it can inhibit NK cell and B cell activation.

The observed effects are facilitated through the stimulation of adrenergic receptors, leading to the secretion of cytokines and chemokines that contribute to the impairment of immune cell function [37]. In conclusion, adrenergic stress plays a key role in the development of the TME by promoting the recruitment and activation of immune-regulatory cells, inhibiting the function of pro-inflammatory immune cells (Figure 1), endorsing the production of immunosuppressive cytokines and chemokines, and promoting tumor angiogenesis [38]. In this review, we discuss the role of adrenergic stress, specifically in T lymphocytes.

## 4. Eminent Role of β2-AR in T Cells

Catecholamines can activate five distinct types of adrenergic receptors. Some of them have been reported to be expressed in immune cells. T lymphocytes express both α-AR and β-AR receptors [39,40]. The expression of α-AR on T lymphocytes has remained controversial, and there are various conflicting data on this issue. There are some studies which have completely denied the possibility of the presence of α-AR in T cell populations, while others have reported the expression of α-AR in lymphocyte populations upon LPS stimulation [41,42]. The conflict in these studies might be due to lack of availability of specific antibodies. The data regarding α-AR expression mostly rely on RT-PCR analysis, which is prone to contamination. Whatever the reason might be, the fact is that evidence regarding the role of α-AR in T lymphocytes is insufficient and less documented [43]. On the other hand, the role of β2-AR in T lymphocytes is well documented and has been the focus of much recent research [21,44,45].

The effects of NA on CD8+ T cells are primarily mediated by β2-adrenergic receptors (β2-AR) rather than other receptors of the same class. It was found that the α-AR antagonist phentolamine and the β1-AR-specific antagonist atenolol were unable to reverse the effects of NA in CD8+ T cells. This was demonstrated by the fact that phentolamine did not alter the expression of IL-1 and IL-6 genes under the influence of NA. Therefore, this study suggests that β2-AR plays a more exclusive role in the suppression of T cell receptor (TCR)-mediated human and mouse CD8+ T cell effector functions [46,47]. Moreover, the β2-AR specific blocker nadolol can abolish NA-mediated effects on CD8+ T cells. On the other hand, β2-AR-specific agonist terbutaline mimics the inhibitory actions of NA [47]. In a model of influenza virus, the role of beta-adrenergic receptors was assessed by pharmacological inhibition of β2-AR as well as chemical inhibition of SNS. Both of the aforementioned systems mimicked each other, signifying the importance and vitality of β2-AR signaling in T cell populations [48].

Pharmacological inhibition or genetic deletion of β2-AR in mice can improve anti-tumor immunity in mice models of colon and breast cancer under stress conditions. Inhibition of β2-AR signaling led to an improved number and function of CD8+ T cells, owing to better tumor control as compared to the control group [49]. The beneficial effects of β2-AR blockade (pharmacological or genetic) in the case of cancer depend upon CD8+ T cells [49,50]. CD8+ T cells, being the backbone of CAR-T therapy, are vital for anti-tumor immune responses. Certain studies have demonstrated that blocking β2-AR signaling can halt the growth of tumors. The mechanism of this phenomenon is relatively complex, and often, intrinsic signaling pathways of cancer cells are held responsible for adrenergic blockade-mediated anti-tumor responses. The poor prognosis of cancer might be due to adrenergic stress-driven loss of T cell activity. In the absence of CD8+ T cells, propranolol and β2-AR knockout (KO) mice do not exert their anti-tumor effects, which signifies that the β2-AR blockade, apart from affecting tumor intrinsic pathways, improves the status of anti-tumor immunity and, hence, leads to better anti-tumor control [51]. Overall, the evidence suggests that the inhibitory effects of NA on T cells are mainly due to the exclusive involvement of β2-AR, highlighting the importance of this receptor subtype in regulating immune responses.

## 5. β2-AR and T Cell Signaling

The molecular mechanism of β2-AR is elusive, as it works with different transducers (Gs, Gi, and β-Arrestin) and exhibits different intracellular signaling pathways. Continuous stimulation of nor-adrenaline in TME triggers the canonical pathway of β2-AR expressed in T lymphocytes [29]. Once nor-adrenaline binds to its receptor, it leads to uncoupling of the Gα subunit from the other two (β and γ) subunits, along with the conversion of GDP to GTP. The Gαs-GTP complex is recruited to lipid rafts, where the activation of adenyl cyclase (AC) takes place. Adenyl cyclase catalyzes the ATP and converts it into cyclic-AMP (cAMP), which further activates protein kinase A (PKA). cAMP can activate both isoforms of PKA i.e., PKA-1 and PKA-2, but only PKA-1 has a recognized role in T cell signaling [52]. In the context of CAR-T, cAMP-dependent PKA-1 phosphorylation activates C-terminal SRC kinase (CSK), which phosphorylates and inactivates Lck. Lymphocyte-specific protein tyrosine kinase (LcK) is important for the activation and phosphorylation of CD3-zeta chain-associated protein kinase 70 (ZAP70) and other substrates that initiate downstream signaling, such as AP-1, NFAT, Erk, and NF-kB [53]. These transcription factors play an important role in T and CAR-T cell activation, proliferation, survival, and cytotoxic ability. β2-AR inhibits ZAP70 activation and CD-3ζ tonnic signaling by activating cAMP and PKA (Figure 2). On the other hand, β2-AR upregulates the expression of checkpoint receptor PD-1, which further inhibits CD28-mediated T cell activation signaling [50]. Additionally, the adrenergic system can modulate T cell apoptosis in a PKA-independent manner by engaging the Src family tyrosine kinase Lck [54]. Overall, β2-AR signaling in T cells is complex, and more studies are needed in order to unveil the molecular mechanisms in detail.

## 6. β2-AR as a Potential Target for Enhancing Cancer Immunotherapy Efficacy

### 6.1. β2-AR and T Cell Co-Stimulation

Co-stimulation is a critical step in the activation of T cells, which requires not only binding of the T cell receptor (TCR) to the antigen presented by DCs, but also the engagement of co-stimulatory molecules such as CD28, 41-BB, etc., on the surfaces of T cells [55]. In many solid tumors, abnormal cell activation has been observed in both mouse and human populations. This is because tumor-infiltrating lymphocytes (TILs) demonstrate a decrease in CD-3ζ tonic signaling. Defects in TCR signaling, in turn, affect the proliferative as well as cytotoxic ability [56]. Some studies have shown that β2-AR signaling can inhibit co-stimulation by downregulating the expression of CD28 on the surfaces of T cells. For example, by employing human T cells that have undergone purification, an experiment was conducted to evaluate of the proliferative potential following stimulation with immobilized anti-CD3 monoclonal antibody (mAb) while under the influence of the beta-adrenergic agonist isoproterenol (ISO). The proliferative capacity of T cells, including their CD4+, CD8+, and CD45RO+ subsets, in response to anti-CD3 mAb was suppressed in a manner that was dependent on the dosage of ISO [51,57].

For CAR-T therapy to magnificently eliminate existing tumors, novel strategies are being employed to boost co-stimulatory signals specifically within the tumor microenvironment. The introduction of “switch receptors” and “inverted cytokine receptors” (ICRs) are examples of countless efforts to boost T cell co-stimulation and, hence, augment the immunotherapeutic response to CAR-T cell therapy [58]. The mechanism by which β2-AR signaling regulates T cell co-stimulation has been demonstrated in Figure 2. One proposed mechanism is that β2-AR signaling can inhibit the activation of NF-κB, a transcription factor that is critical for the expression of co-stimulatory molecules [38,59]. One potential strategy to boost CAR-T cell co-stimulation could be to inhibit β2-AR.

### 6.2. β2-AR and T Cell Activation Signals

T cells require two to three signals in order to exhibit complete activation. Antigen-presenting cells or cancer cells present antigen (MHC-1) on their surfaces, which binds with T cell receptor TCR and give the first signal for activation. Co-stimulatory molecules, such as CD28 and 4-1BB, bind with their ligands, B7.1 or B7.2, on APCs, providing a crucial “second signal” (Figure 3) [60,61]. Soon after activation, cells undergo extensive proliferation and release pro-inflammatory cytokines such as IL-2, IL-7, IL-15, and IL-21, which bind to their receptors on T cell surfaces in a positive feedback loop manner, providing the third signal. Extensive proliferation is important, as a significantly greater number of T cells are required in order to successfully inhibit the rapidly growing tumor tissue [62]. Although CAR-T has had enormous success in many tumors with excellent remission rates, still there are reports of relapse. There can be several explanations for this failure of treatment by CAR-T therapy, including adrenergic stress-mediated inhibition of T cell activation and, subsequently, proliferation at the tumor site [44,63,64].

CD69 is an early-activation marker expressed on various immune cells, including T cells. Its expression is rapidly induced upon T cell activation and it is involved in the regulation of T cell activation, differentiation, and effector functions [65,66]. Some studies have suggested that β2-AR can inhibit the activation marker CD69 in T cells [47,49,67,68]. For example, when exposed to isoproterenol (ISO), a beta-2 adrenergic receptor agonist, CD8+ T-cells showed a significant reduction in CD69 expression both 24 h and 48 h after activation [51,67]. Many recent clinical trials in CAR-T therapy have also demonstrated failure to achieve optimum proliferation of T cells in different tumor models with complex TMEs. Inhibition of proliferation by β2-AR might be due to the fact that it does not allow T cells to activate properly, and subsequently results in decreased proliferation.

### 6.3. β2-AR Self-Regulates T Cell Activation

DCs are professional antigen-presenting cells that act as bridges between the innate and adaptive immune systems [69]. Certain studies have demonstrated that DCs have an excellent ability to penetrate tumor sites, arrest tumor antigens, and subsequently present those antigens to CD8+ T cells after processing; thus, they play vital role in anti-tumor immunity [70]. β2-AR interferes with the anti-presenting ability of DCs, which has been demonstrated to be critical for the proliferation of CD8+ T cells under adrenergic stress in a mouse model of the influenza virus [8,48,71]. As DCs have an extensive ability to prime T lymphocytes, this may possibly have broader implications in immunotherapy. β2-AR inhibits the TLR-4-mediated maturation and antigen-presenting ability of DCs, yet the direct effects of adrenergic signaling in T cells are sufficient to suppress activation as well as proliferation in CD8+ T cells independently of DCs [72]. Recently, Daher Clara et al., 2019, demonstrated that NA-mediated inhibition of proliferation in T cells was not due to the involvement of DCs. Rather, the proliferative capacity and responsiveness of T cells to adrenaline depend upon the activation status [73]. This is why Tns are not susceptible to the effects of NA, but the opposite scenario occurs in effector T cells. This can further be explained by another study, where it was reported that the β2-AR expression is related to the activation status of T cells [46]. Moreover, β2-AR-mediated inhibitory signals are enough to suppress T cell functions independently of their presence and signaling in another important set of immune cells, i.e., DCs.

### 6.4. β2-AR and IL-2 Signaling

Ex vivo expansion of CAR-T cells is an important step prior to the administration of therapy to patients. T cell cultures are supplemented with exogenous cytokines in order to provide proliferative and growth signals. IL-2 is a commonly used cytokine for ex vivo CAR-T expansion [74]. IL-2 resides upstream as well as downstream of Akt molecules in a positive feedback cascade. IL-2 activates the JAK/STAT and JUN pathways, both of which are involved in the production of growth factors required for T cell proliferation [75]. β2-AR has been shown to inhibit IL-2 signaling in T cells [76]. The β2-AR signaling pathway appears to have a multifaceted role in the regulation of lymphoid cell proliferation and survival. Additionally, β2-AR activation induces threonine phosphorylation of the IL-2Rβ, indicating a potential mechanism for ISO-induced changes in lymphoid cell function. In contrast, when cells were treated with ISO before IL-2 stimulation, an inhibitory signal was generated, which hindered IL-2-induced activation of the JAK/STAT, MEK/ERK, and PI3K pathways. This inhibition was attributed to disruption of the IL-2R beta-gamma chain complex, which is necessary for downstream signaling and subsequent cell proliferation [77].

### 6.5. Metabolic Profiling of T Lymphocytes

During the resting phase, T cells do not have strict requirements, as the demand for nutrients is low. So, glucose is metabolized through oxidative phosphorylation. Metabolic reprogramming in T cells refers to the changes in the metabolic pathways that occur when T cells are activated to perform their immune functions [78]. Activated T cells undergo a shift from oxidative phosphorylation to glycolysis, a process known as the Warburg effect. This shift in metabolism allows T cells to rapidly produce the energy and biosynthetic intermediates required for the production of cytokines and the proliferation of effector T cells. This metabolic reprogramming is regulated by a variety of signaling pathways and transcription factors, including mTOR, HIF-1α, and c-Myc [79,80].

#### 6.5.1. Role of β2-AR in T Cell Glycolysis

Interestingly, different subsets of T cells exhibit distinct metabolic profiles. For example, effector CD8+ T cells primarily rely on glycolysis. Activated T cells increase their glucose uptake and upregulate the expression of glycolytic enzymes, such as hexokinase and pyruvate kinase, to support glycolysis. This increased glycolytic activity is crucial for T cell activation and function. β2-AR has been reported to downregulate the expression of the GLUT-1 gene and glycolytic enzymes in T cells during the T cell activation phase [51]. The authors demonstrated that ISO treatment leads to a reduced expression of glucose transporter 1 (GLUT-1) following activation and a subsequent decrease in glucose uptake and glycolysis in comparison to CD8+ T-cells activated in the absence of ISO. Importantly, this effect was specific to β2-AR signaling, as it was not observed in CD8+ T cells deficient in β2-AR and could be blocked by the β-AR antagonist propranolol. This is why, in another study, knockdown of β2-AR in mice led to increased glycolysis and oxidative phosphorylation in T cells [51,81]. This means that the adrenergic stress inside the tumor microenvironment can negatively impact the metabolic profile of CAR-T cells, and that any intervention for blocking β2-AR in T lymphocytes might be a potential strategy to boost the efficacy of CAR-T cell therapy. Increased glycolytic activity by inhibiting β2-AR signaling might be due to the increased expression of co-stimulatory molecules and decreased expression of co-inhibitory molecules on the surfaces of tumor-infiltrating lymphocytes [52,53]. Inhibiting glycolysis might lead to reduced T cell proliferation and cytokine release, rendering T cells dysfunctional against cancers [82].

Hypoxia-inducible factor 1 alpha (HIF-1α) is a pivotal transcription factor governing the expression of numerous genes associated with glucose metabolism, among which is GLUT-1. β2-AR signaling has been shown to inhibit HIF-1α expression in CD8+ T cells, which could further lead to reduced GLUT-1 expression [83]. This may have important implications in the determination of immunotherapeutic efficacy.

#### 6.5.2. Role of β2-AR in Mitochondrial Metabolism of T Cells

During the process of T cell activation, the cells upregulate glycolysis, but at the same time, mitochondrial metabolism remains a vibrant part of T cell metabolism. The very reason for this might be that T cells also upregulate mitochondrial respiration during the processes of activation, proliferation, and differentiation [84]. The 4I-BB co-stimulatory domain of CAR-T cells owes to their improved mitochondrial respiration and biogenesis, while, on the other hand, CD28 increases glycolysis and, hence, the effector function of CAR-T cells. Third-generation CAR-T may contain two co-stimulatory domains, which suggests the vitality of the mitochondrial metabolism in CAR-T functioning while dealing with solid tumors [78]. The TME of many solid tumors render T cells exhausted and dysfunctional, which is attributed to a loss of mitochondrial function [85]. In the mitochondria, β2-AR can modulate mitochondrial biogenesis, respiration capacity, and membrane potential [51,67,86]. β2-AR leads to an exhausted phenotype of T cells expressing greater amounts of checkpoint inhibitors, for example, PD-1 [49].

Upregulation of checkpoint inhibitors, especially PD-1, has been shown to negatively regulate the metabolic fitness of T lymphocytes. PD-1 carries its immunosuppressive function by inhibiting PGC1-α expression, which is the master regulator of T cell metabolism. As a result, T lymphocytes are unable to metabolically reprogram themselves in vivo during their infiltration into the tumor milieu in a PD-1/mTOR dependent manner [87,88]. Blocking β2-AR signaling results in a tumor-infiltrating T cell phenotype that is more activated and less exhausted [67,89]. In a study, propranolol-treated mice showed increased mitochondrial mass in CD8+ T cells and CD4+ T cells in vivo. Moreover, blocking β2-AR signaling led to an increased mitochondrial mass and spare respiratory capacity of propranolol-treated tumor-bearing mice by blocking the expression of checkpoint inhibitors on T cells [67]. Overall, the inhibition of the mitochondrial metabolism by β2-AR signaling in CD8+ T cells is thought to contribute to the suppression of T cell activation and function (Figure 4).

### 6.6. β2-AR Mediated Redox Signaling in T Cells

Within the body’s very own cells, a treacherous threat lurks in the form of highly reactive and unstable molecules known as ROS. These harmful agents possess lone, unpaired electrons in their outermost shells, rendering them volatile and prone to uncontrolled behavior. They are not mere invaders, but rather insidious byproducts of metabolic processes, waiting to wreak havoc on T lymphocytes [90]. CAR-T therapy was designed to inhibit oxidative stress. A study demonstrated that CAR-T cells co-expressing catalase had better anti-tumor control and free radical scavenging capacity than control cells [91].

As discussed above, β2-AR can cause metabolic insufficiencies in T cells, and these changes, especially in mitochondrial bioenergetics, can lead to increased ROS production and signaling in T cells [92]. This was demonstrated in a study, where it was shown that nor-epinephrine can lead to the production of excess ROS in T lymphocytes by disrupting the mitochondrial metabolism and, hence, depicting the mitochondria as a major source of ROS in T cells [93]. Some recent research has highlighted that most of the NA-mediated effects in T cells occur via β2-AR receptors, so one of the proposed mechanisms by which β2-AR can alter T cell proliferation and cytokine profile might be their ability to produce excess ROS by disrupting the mitochondrial metabolism [51,67].

## 7. Perspective for CAR-T Cell Therapy

CAR-T cell therapy has gained excellent success in treating cancers and has evolved as a promising new strategy to control tumor growth in patients, especially in blood cancers. However, many cancers, especially those characterized by solid tumors, are mostly resistant to beneficial effects resulting in lower remission rates and relapse [94,95]. Immunotherapies, including CAR-T cell therapy, rely on the patient’s immune system in order to fight the cancer, displaying a robust anti-tumor response which is hindered by immunosuppressive factors within the tumor milieu. Therefore, the need to develop novel therapeutic strategies to reverse immunosuppression never ceases.

Adrenergic stress is one of several culprits behind immunosuppression inside the TME, and has evolved recently as an important regulator of inflammation, immunity, and cancer [96]. Chronic release of NA inside the TME influences tumor growth by impeding immune responses, as is evident in preclinical models of some cancers [10,97]. β2-AR, among adrenergic receptors, is the most studied receptor for its immunosuppressive function in T lymphocytes. β2-AR has been shown to decrease T cell activation, proliferation, and cytokine release in several studies. The receptor performs this function by distorting the mitochondrial fitness of the cell [7,44]. This is why β-antagonists have demonstrated reduced tumor growth in many cancers by increasing the immune response. This has been implicated in checkpoint inhibitor therapies, where β-blocker-mediated inhibition of β2 signaling led to significant improvement in response to checkpoint inhibitor therapies. Blocking β2 signaling has been shown to increase tumor-infiltrating CD8+ T cells, and hence can potentiate immunotherapy [98]. Thus, in the light of the present literature, we propose that blocking β2-adrenergic signaling in T cells can be an applicable strategy to improve CAR-T cell functioning in complex tumor microenvironments.

## 8. Concluding Remarks

The inhibitory role of β2-AR in T lymphocytes is often demonstrated, providing sufficient rationale to translate this research into CAR-T research by interfering with adrenergic signaling for the benefit of the immunotherapeutic response. Implying current knowledge about the role of β2-AR in tumor microenvironments, and especially how it impacts T cell functions, can synergize the anti-tumor efficacy of CAR-T cell therapies in future clinical trials. More studies unearthing the molecular mechanisms of β2-AR-mediated T cell inhibition will pave the way for the design of novel immunotherapies for cancer treatment.

## Figures and Tables

**Figure 1 ijms-24-12837-f001:**
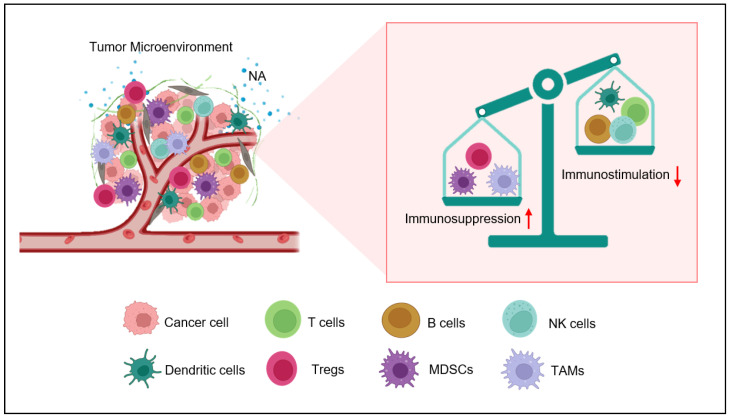
Nor-adrenaline (NA)-mediated immunosuppression inside the tumor microenvironment. Nor-adrenaline released in the tumor microenvironment activates β2-AR on the surfaces of various immune cells. The activation of these adrenergic receptors activates myeloid-derived suppresser cells (MDSCs) and recruits T-regulatory cells (Tregs) and M2 tumor-associated macrophages (TAMS). On the other hand, the activity of natural killer (NK) cells, T cells, and dendritic cells (DCs) is inhibited by β2-AR signaling.

**Figure 2 ijms-24-12837-f002:**
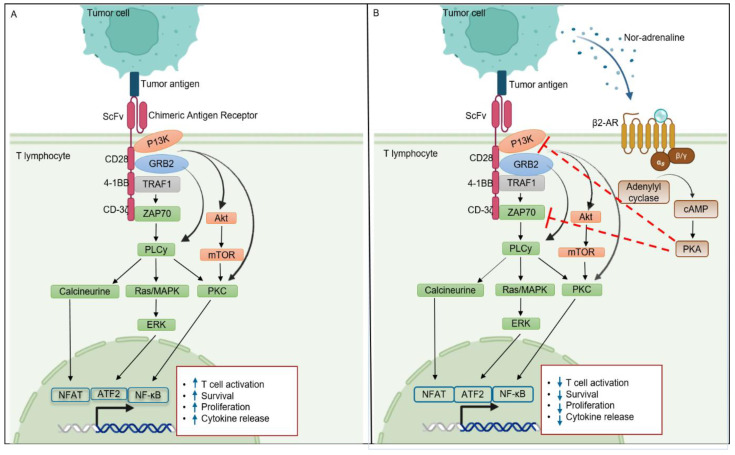
β2-AR mediated CAR-T cell signaling. Panel (**A**) indicates the signaling pathways of CAR-T cells after activation. Panel (**B**) indicates that the activation of β2-AR hinders the normal functioning of CAR-T cells by inhibiting the phosphorylation of ZAP70 in a PKA-dependent manner. As a result, various downstream signaling pathways (NFAT, ERK, ATF2, and NF-κB) that regulate T cell function are also inhibited. Black and blue colored arrows represent activation while red colored dotted arrows represents inhibition.

**Figure 3 ijms-24-12837-f003:**
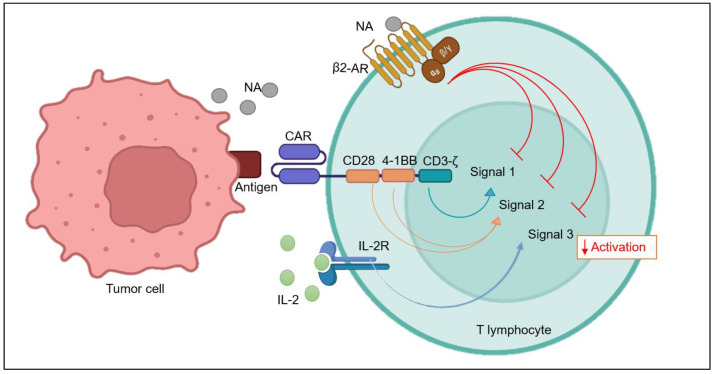
Schematic representation of β2-AR-mediated inhibition of CAR-T cell function. Optimum CAR-T activation requires signaling via all three signals (Signal 1, Signal 2, and Signal 3). Adrenergic signaling via β2-AR activation on the surfaces of T cells can block CAR-T cell activation signaling by inhibiting the stimulatory domain, the co-stimulatory domain, and the activity of IL-2 receptors. Red arrow depicts inhibition.

**Figure 4 ijms-24-12837-f004:**
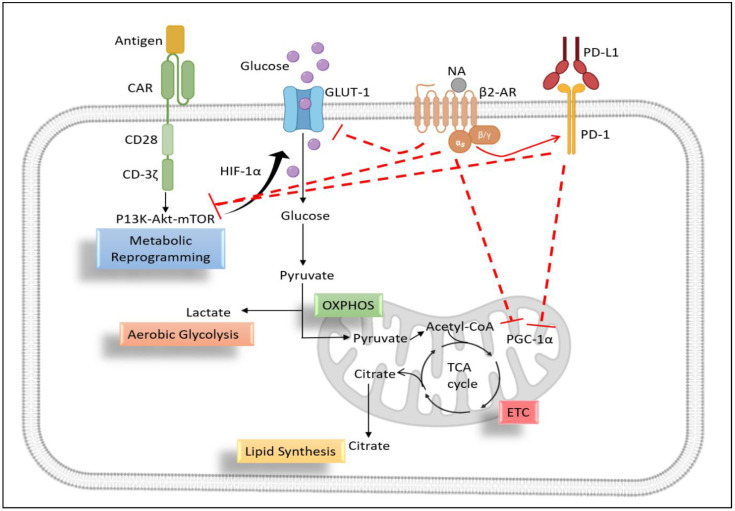
Proposed role of β2-adrenergic signaling in metabolic reprogramming of CAR-T cells. CAR-T cells have been activated after recognizing their specific antigen on tumor cells. CD3-ζ and co-stimulatory molecules undergo metabolic reprogramming by activation of the P13K-Akt-mTOR pathway. This pathway increases the expression of GLUT-1 and release pyruvate as a by-product of aerobic glycolysis. Pyruvate enters the mitochondria and begins the TCA cycle, which, in turn, produces a high amount of energy for T cell functioning. Nor-adrenaline in the TME binds with β2-AR and releases PKA, which can inhibit GLUT-1 and CD-3ζ, CD28, and 4-1BB-mediated activation signaling, and therefore blocks metabolic reprogramming. On the other hand, β2-AR increases the inhibitor (PD-1) signaling in T cells, which also hinders the CAR signaling and leads to mitochondrial dysfunction.

## Data Availability

Not applicable.

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
