# Peer review of "β2-Adrenergic Receptor Mediated Inhibition of T Cell Function and Its Implications for CAR-T Cell Therapy"

_ijms, 2023, doi:10.3390/ijms241612837_

Round 1

Reviewer 1 Report

The review article titled β2-Adrenergic Receptor Mediated Inhibition of T Cell Function And Its Implications For CAR-T Cell Therapy by Farooq et al. is addressing an interesting subject. The manuscript is organized and contains adequate number of relevant articles to support the ideas.

Major concern: The authors need to elaborate on the apoptotic proteins being modulated that would affect CAR T cell functionality under section 5. β2-. AR and T cell signaling where the authors have discussions on signaling pathways. 

 English editing is needed for this article. 

Author Response

We are deeply grateful for the time and effort you have dedicated to review our manuscript. Many Thanks for the encouraging comments and overall positive feedback on our work.

Point-1: The authors need to elaborate on the apoptotic proteins being modulated that would affect CAR T cell functionality under section 5. β2-AR and T cell signaling where the authors have discussions on signaling pathways.

Response; As β2-AR distribution in body systems is diverse, so is their function. In certain cells like Cardiomyocytes, activation of β2-AR leads to resistance to apoptosis. In T lymphocytes, Briefly, activation of these receptors in T cells leads to apoptosis in PKA-independent fashion. The new information along with reference has been incorporated in the relevant section as advised. The changes have been highlighted in the relevant section.

Point 2: English editing is needed

Response; The comment has been addressed. English language has been improved. Some sentences with complex structure were simplified for the ease of reader. Paper was carefully read for spellings, punctuations and grammatical mistakes, and much more polished manuscript has been uploaded. The changes were highlighted in relevant section.

Reviewer 2 Report

The authors focused on the β2-Adrenergic Receptor Mediated Inhibition of T Cell Function And Its Implications For CAR-T Cell Therapy. Please see my main suggestions below:

Remove the figure from the Abstract. You should provide a Graphical Abstract, separately than the main manuscript.

Check the Instructions for authors regarding Acronyms/Abbreviations/Initialisms, which should be defined the first time they appear in each of three sections: the abstract; the main text; the first figure or table. When defined for the first time, the acronym/abbreviation/initialism should be added in parentheses after the written-out form. Apply to the main text and under figures/tables, explaining all abbreviations used in all those figures/tables.

Better quality figures are required.

Section 6. Provide a table summarizing most significant studies in the topic, for all the sections you detailed.

Subsections 6.1. to 6.4. Correct Editing for the title of these subsections

Last section should be Conclusions. What novelty and/or special aspects brings your research to the topic.

Check your editing for the entire manuscript using Editing Tool for Word.

References should be in the MDPI style, providing all the information requested by the Instructions for Authors - please check them.

Good English. Minor editing errors.

Author Response

We are thankful to you for sparing your precious time to review our paper and provide us insightful feedback. Here is point by point response.

Point 1: Remove the figure from the Abstract. You should provide a Graphical Abstract, separately than the main manuscript

Response; The graphical abstract has been removed from the main-manuscript and is supplemented separately.

Point 2: Check the Instructions for authors regarding Acronyms/Abbreviations/Initialisms, which should be defined the first time they appear in each of three sections: the abstract; the main text; the first figure or table. When defined for the first time, the acronym/abbreviation/initialism should be added in parentheses after the written-out form. Apply to the main text and under figures/tables, explaining all abbreviations used in all those figures/tables.

Response: We appreciate for very valid concern raised by worthy reviewer. There were some unexplained abbreviations in the text. They have been rectified accordingly.

Point 3: Better quality figures are required.

Reponse: Thanks for your suggestion. The figures were uploaded as JPEG files with clear and readable images. Sometimes the figures mislay their pixels while converting into pdfs. We think this was the reason.

Point 4: Section 6. Provide a table summarizing most significant studies in the topic, for all the sections you detailed.

Response: As far β2-AR, studies in CAR-T therapies are yet to be published (as per our knowledge). So, the section 6 was designed by supplementing information about role of β2-AR in T-cells, parallel to working of Chimeric antigen receptor. As there are no direct studies, so we tried to extrapolate the available data in the context of CAR-T therapies to give a new perspective. Due to availability of a little or no data, the table was not possible relevant to this part. As our major focus was CAR-T cell therapy, so the proposed mechanisms were illustrated for reader clarification.

Point 5: Subsections 6.1. to 6.4. Correct Editing for the title of these subsections

Response: The comment has been addressed

Point 6: Last section should be Conclusions. What novelty and/or special aspects brings your research to the topic.

Response: The comment was addressed. The conclusions were added.

Point 7: Check your editing for the entire manuscript using Editing Tool for Word. References should be in the MDPI style, providing all the information requested by the Instructions for Authors - please check them.

Response: The entire manuscript was proofread. There were many minor English language mistakes that were removed. Some sentenced were rephrased to make them more comprehendible for readers. All the references were formatted according to Journal guidelines. 

Round 2

Reviewer 2 Report

the paper was properly improved.